



# Hydrometeorological, glaciological and geospatial research data from the Peyto Glacier Research Basin in the Canadian Rockies

Dhiraj Pradhananga[1,2,3], John W. Pomeroy[1], Caroline Aubry-Wake[1], D. Scott Munro[1,4], Joseph
Shea[1,5], Michael N. Demuth[1,6,7], Nammy Hang Kirat[3], Brian Menounos[5,8], Kriti Mukherjee[5]

[1] Centre for Hydrology, University of Saskatchewan, 101-121 Research Drive, Saskatoon, SK S7N 1K2, Canada

[2] Department of Meteorology, Tri-Chandra Multiple Campus, Tribhuvan University, Kathmandu, Nepal

[3] The Small Earth Nepal, PO Box 20533, Kathmandu, Nepal

[4] Department of Geography, University of Toronto Mississauga, Ontario, L5L 1C6, Canada

[5] Geography Program, University of Northern British Columbia, 3333 University Way, Prince George, BC V2N 4Z9,
Canada

[6] Geological Survey of Canada, Natural Resources Canada, Ottawa, ON 601 Booth St, Ottawa, ON K1A 0E8, Canada

[7] University of Victoria, Victoria, Canada, 3800 Finnerty Road, Victoria, BC V8P 5C2

[8] Natural Resources and Environmental Studies Institute, University of Northern British Columbia

*Correspondence to:* Dhiraj Pradhananga (dhiraj.pradhananga@usask.ca)

**Abstract.** This paper presents hydrometeorological, glaciological and geospatial data of the Peyto Glacier Research
Basin (PGRB) in the Canadian Rockies. Peyto Glacier has been of interest to glaciological and hydrological
researchers since the 1960s, when it was chosen as one of five glacier basins in Canada for the study of mass and
water balance during the International Hydrological Decade (IHD, 1965-1974). Intensive studies of the glacier and
observations of the glacier mass balance continued after the IHD, when the initial seasonal meteorological stations
were discontinued, then restarted as continuous stations in the late 1980s. The corresponding hydrometric observations
were discontinued in 1977 and restarted in 2013. Data sets presented in this paper include: high resolution, co-
registered DEMs derived from original air photos and LiDAR surveys; hourly off-glacier meteorological data recorded
from 1987 to present; precipitation data from nearby Bow Summit; and long-term hydrological and glaciological
model forcing datasets derived from bias-corrected reanalysis products. These data are crucial for studying climate
change and variability in the basin, and to understanding the hydrological responses of the basin to both glacier and
climate change.  The comprehensive data set for the PGRB is a valuable and exceptionally long-standing testament to
the impacts of climate change on the cryosphere in the high mountain environment. The dataset is publicly available
from Federated Research Data Repository at https://doi.org/10.20383/101.0259 (Pradhananga et al., 2020).



# 1 Introduction

Peyto Glacier (Figure 1) is in Banff National Park, Alberta, Canada. It forms part of the Wapta Icefield in the Waputik Range. The Wapta Icefield is one of the southernmost icefield complexes of the Canadian Rocky Mountains and is a high mountain headwater for the Columbia and Saskatchewan-Nelson river systems in western Canada. Peyto Glacier

contributes runoff to the Mistaya River Basin, a headwater of the North Saskatchewan River, which eventually reaches Hudson Bay via the Nelson River. Glaciers and snowpacks in these headwater systems are important sources of water that support industry, agriculture, hydropower generation, drinking water and the environment. The meltwater from this glacier and others in the region is a crucial component of streamflow during dry late summer periods (Comeau et al., 2009; Demuth et al., 2008; Hopkinson and Young, 1998).

The first geophysical record of Peyto Glacier goes back to a photograph taken by Walter D. Wilcox in 1896, followed by subsequent photographs and a map from the Alberta-British Columbia Interprovincial Boundary Commission Survey (Tennant and Menounos, 2013). Significant research on the glacier began in 1965 when it was selected as one of the research sites for the International Hydrological Decade (IHD). The scope and observational resources have

varied since then, with more recent advances and restoration of observations (Munro, 2013). Mass balance observations continued after the IHD, but discharge observations ended in 1977. The stream gauge site was washed away by a flood in July 1983. Discharge measurements resumed in 2013, recorded by the Centre for Hydrology at the University of Saskatchewan (USask) at a new gauging site located 1.5 km upstream from the previous location. A year-round automatic weather station, operating since 1987 (Munro, 2013), was upgraded in 2013 as part of the

Canadian Rockies Hydrological Observatory observation system and is now operated by USask.

Collecting continuous, high-quality data from remote and difficult-to-access alpine glacier basins can be a challenge. Lafrenière and Sharp (2003) and Rasouli et al. (2018), for example, noted the impact of power source failures on automatic weather station (AWS) records, such as to cause significant data gaps. High snow accumulations during

winter can bury an AWS installed on the glacier surface, and riming can compromise instrument performance; in turn, high summer melt can cause stations to tilt, or fall over. Climate data availability and accuracy in the Peyto Glacier Research Basin (PGRB) suffer from many such irregularities. Therefore, affected data must be infilled or corrected before they can be used for medium and long-term studies.

The World Glacier Monitoring Service (WGMS) has listed Peyto Glacier as a 'reference glacier' for mass balance, in consideration of its mass balance data record of over 50 years. Peyto Glacier is also one of the observing sites operated by the Geological Survey of Canada's Glacier-Climate Observing Program (Demuth and Ednie, 2016). Therefore, the PGRB can be considered an outdoor laboratory for conducting hydrological research, as proposed by Seyfried (2003); however, a single document that describes the relevant hydro-meteorological datasets is needed. This paper details the

meteorological forcing data that were created for driving hydrological models of Peyto Glacier, along with related hydrological and geospatial datasets used for model evaluation, mainly for three time periods: 1965-1974, 1987-2012 and 2013-2018. These datasets include historical archived data from the IHD period and recent data from both on-ice

and off-ice stations.  Glaciological mass balance measurements, using ablation stakes and snow pits, have been carried out continuously since the beginning of the IHD period, and a comprehensive account of the first 14 years of mass balance results appeared in Young (1981). Mass balance data reported from Peyto Glacier have been used by many researchers (Bitz and Battisti, 1999; Demuth et al., 2008; Demuth and Keller, 2006; Letréguilly, 1988; Letréguilly and
5    Reynaud, 1989; Marshall et al., 2011; Matulla et al., 2009; Menounos et al., 2019; Østrem, 1973; Schiefer et al., 2007; Shea and Marshall, 2007; Watson et al., 2006; Watson and Luckman, 2004; Zemp et al., 2015) as reference data for the region, but the collection of data that could be used for modeling purposes has never been assembled in a single description until now.

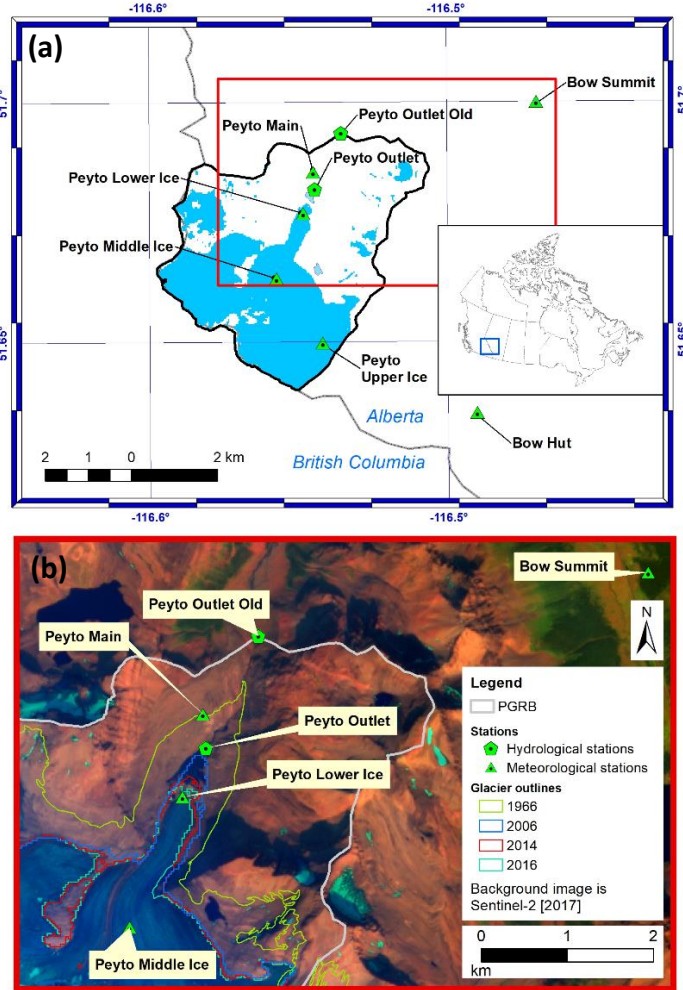

10    **Figure 1: Peyto Glacier Research Basin (PBRB). (a) locations of PGRB and the hydro-meteorological stations, (b) past and present glacier extents.**

## 2   Peyto Glacier Research Basin (PGRB)

The PGRB is in the Canadian Rockies, on the eastern side of the Continental Divide, at latitude 51.67 N and longitude 116.55 W. This heavily glacierized basin is 23.6 km$^2$ in area, ranging in elevation from 1907 to 3152 m. It is located in a predominantly sedimentary geological region, with surrounding mountains formed from hard, resistant dolomite
(Young and Stanley, 1976). The basin has been well monitored over a 50-year observational period (Shea *et al.*, 2009). During the 1960s, the area of the glacier was 13.4 km$^2$, but it has been continuously losing mass and area since at least the 1920s (Tennant et al., 2012), shrinking to an area of 9.87 km$^2$ as of 2018 (Figure 1). Repeat ground-based photography (Figure 2) from 1902 and 2002 show the glacier retreat that has occurred over the 20$^{th}$ century. A new proglacial lake has since formed at the tongue of the glacier that increases in size every year and has been informally
named 'Lake Munro' by USask to honor D. Scott Munro's research contribution to the glacier basin. Peyto Creek flows out of Lake Munro, draining the PGRB into Peyto Lake, thus supplying water to the Mistaya River.

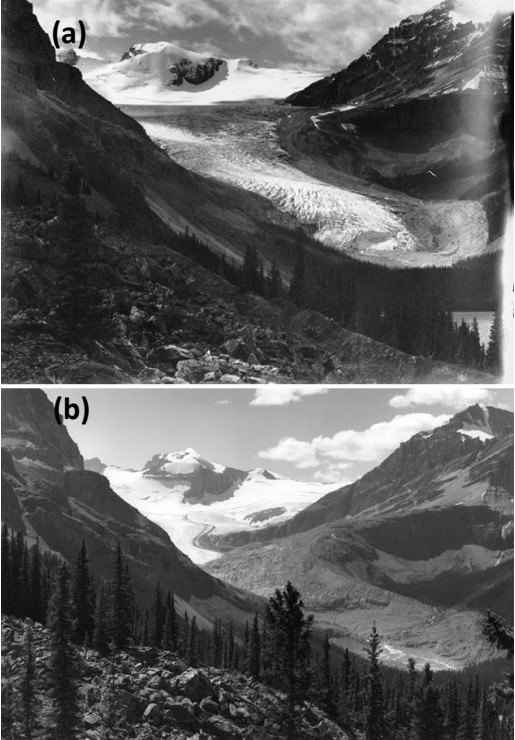

**Figure 2: Peyto Glacier in (a) 1902 (V653/NA-1127, Vaux Family, Whyte Museum of the Canadian Rockies, Whyte.org), and (b) 2002 (courtesy Henry Vaux Jr.).**



## 2.1 Hydrometeorological sites

Meteorological observations were taken over the summer months (June – September) during the IHD at the Peyto Creek Base Station adjacent to the glacier terminus, herein referred to as Peyto Main (Figure 1). After becoming dormant in 1974, the station was re-established at the same location in September 1987. Table 2 and Table 3 detail

the meteorological variables and instruments used to record them during the IHD and the post-IHD period. Three meteorological stations were also established on the glacier surface for post-IHD micrometeorological studies by D. Scott Munro in different elevation zones: Lower, Middle and Upper ice stations. These were originally positioned to represent different glacier net mass balance zones – ablation zone, equilibrium line zone, and accumulation zone. Since 2012, USask, has continued these stations with new instruments, but they have been relocated to accommodate

changing glacier geometry and rising elevation of the equilibrium line. These data, however, are not continuous because only the Lower Ice station was maintained after 2013 due to rapid ice melt causing tower collapse and subsequent station burial at the higher elevation sites. Peyto Outlet is a hydrometric station that measures glacier meltwater runoff at the outlet of Lake Munro.

The AWS sites in the PGRB are now a part of the Canadian Rockies Hydrological Observatory (https://research-groups.usask.ca/hydrology/science/research-facilities/crho.php#Overview), a USask network of 35 hydrometeorological and hydrometric stations in the Canadian Rockies. They are also part of the cryospheric surface observation network (CryoNet) of the World Meteorological Organisation Global Cryosphere Watch (WMO-GCW) - http://globalcryospherewatch.org/cryonet. Peyto Main and Peyto Lower Ice are listed as Reference CryoNet stations,

whereas the others are Contributing CryoNet Stations of the GCW. Figure 1 and Table 1 contain the locational information, data collection periods and data elements recorded at the stations, with selected stations shown in Figure 3.

**Table 1: CryoNet station data.**

| Station Name | Station Type[1] | Geographical Coordinates Elevation above sea level | Variables | Data Period |
|---|---|---|---|---|
| Peyto Main | Reference | 51.68549 N; 116.54495 W 2240 m | Ta, RH, Ws, Wd, Ts, Qsi, Qso, Qli, Qlo, Ppt, P, Sd | July 2013 – Aug 2019[2] |
| Peyto Main Old | Reference | 51.68541 N; 116.54467 W 2240 m | Ta, RH, Ws, Wd, Ts, Qsi, Qli, Ppt, P | Sept 1987- July 2018[3,5] |
| Peyto Main IHD | Reference | 51.68549 N; 116.54467 W 2240 m | Ta, RH, Ws, Qsi, Ppt, Sunshine hours | 1965-1974[4] |
| Peyto Lower Ice[7] | Reference | 51.67669 N; 116.53399 W 2173-2183 m | Ta, RH, Ws, Ts, Qsi, Qso, Sd | Aug 1995- Aug 2019[6] |
| Peyto Middle Ice[7] | Contributing | 51.66293 N; 116.55754 W 2454-2461 m | Ta, RH, Ws, Ts, Sd | Sept. 2000- Sept 2013[3] |
| Peyto Upper Ice[7] | Contributing | 51.64930 N; 116.53651 W 2709 m | Ta, RH, Ws, Ts, Sd | July 2000- Sept 2013[3] |
| Bow Hut | Primary | 51.63517 N; 116.49031 W 2421 m | Ta, RH, Ws, Wd, Sd | Oct 2012- Sept 2018[2] |
| Peyto Outlet | Primary | 51.68111 N; 116.54472 W 2150 m | Ta, Runoff | June 2013- Sept 2018[2] |



Ta = air temperature, RH = relative humidity, Ws = wind speed, Wd = wind direction, Ts = soil/snow/firn/ice temperature, Qsi, Qso = incoming and outgoing shortwave radiation, Qli, Qlo incoming and outgoing longwave radiation, Ppt = precipitation, P = air pressure, Sd = snow depth (SR50)

_______________

5 [1]Station type according to CryoNet

[2]recorded at fifteen-minute intervals

[3]recorded hourly until September 2008, at thirty-minute intervals then after

[4]daily data for the summer months

[5]Qli is available, beginning September 1998

10 [6]hourly until September 2008, then at 30-minute intervals to 2015, 15-minute intervals 2015-2019. Qsi and Qso measurements from 2007 to 2008; Qsi measurements available again since 2015

[7]snowpack glacier accumulation and ablation data are also available; Ice stations have several data gaps, mainly in middle and upper ice station records

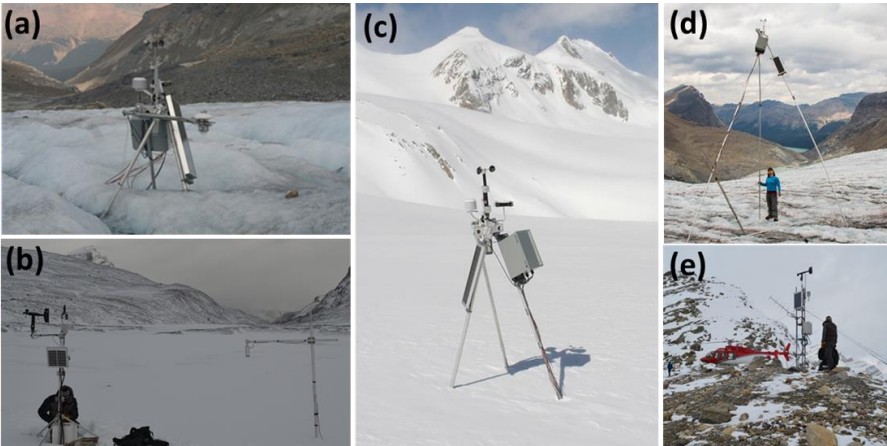

**Figure 3: Photographs of selected CryoNet stations in the PGRB. (a) Peyto Lower Ice (2009), (b) Peyto Lower Ice (Oct 2016), (c) Peyto Middle Ice (April 2006), (d) Peyto Middle Ice (Sept 2015), and (e) Bow Hut (Oct 2016). Photographs by Dhiraj Pradhananga (b & e), D. Scott Munro (a & c), and Angus Duncan (d).**

## 3  Data

Young and Stanley (1976) documented the glaciological and hydro-meteorological data collected within the glacier basin during the IHD. Past studies over the glacier are also well documented in '*Peyto Glacier: One Century of*

25 *Science*' (Demuth et al., 2006), which provides details on the mass balance data until 1995, along with the hypsometry of the glacier.

### 3.1  Meteorological data – historical and present

Young and Stanley (1976) describe meteorological and mass balance data for the period 1965-1974. Air temperature, relative humidity, global radiation, hours of bright sunshine, cloud cover, wind speed, and precipitation were recorded



during the summer months at a meteorological station located in the base camp (Figure 4a) and documented as 'Peyto Creek Base Station' observations. The data collection details and instruments used are described in publications of the Inland Waters Directorate of Environment Canada (Goodison, 1972; Young and Stanley, 1976).

Automatic weather stations were first installed at on- and off-glacier sites for micrometeorological studies and retained for long-term data collection. The data from Peyto Main Old (Figure 4b) are hourly prior to September 2008, and half-hourly thereafter to 2018. The Peyto Main station (Lat: 51.51 N, Long: 123.44 W, Elevation: 2237 m) was installed near Peyto Main Old in 2013, with new instruments and settings (Table 4). Fifteen-minute intervals apply to the data of Peyto Main that were collected from 2013 to 2020. Some data (2002-2007) for Peyto Main Old were published

(Munro, 2011b) in support of the IP3 Network initiative: *Improving Processes & Parameterization for Prediction in Cold Regions Hydrology* (IP3, 2010). The details of the IP3 Network and AWS data from the Peyto Main Old site (Table 2) are available at http://www.usask.ca/ip3/data.php.

**Table 2: Details of hourly PGRB meteorological data referred to in Goodison (1972) and Munro (2011b).**

| Variables | Instruments | |
|---|---|---|
| | **Peyto Main Old** | **Peyto Main IHD (June – August)** |
| Air temperature and relative humidity | Campbell Model 207/Vaisala HMP35, YSI[1] thermistor | Lambrecht 252 Thermo-Hygrograph, CMS[2] max. and min. thermometers |
| Ground/snow temperatures | YSI thermistor | N/A |
| Wind speed and direction | RM Young anemometer & vane | MK II totalizing anemometer |
| Precipitation | Recording gauge[3], CMS tipping bucket | Pluvius /CMS 3" rain gauge |
| Sunshine hours | | Campbell-Stokes sunshine recorder |
| Incoming longwave radiation | Epply PIR pyrgeometer | |
| Incoming shortwave radiation | Kipp & Zonen CMP 6/11 pyranometer | Belfort 5-3850 pyranograph |

⎯⎯⎯⎯⎯⎯⎯⎯⎯⎯⎯

[1]YSI stands for Yellow Springs Instruments.
[2]CMS stands for Canadian Meteorological Service (now MSC, the Meteorological Service of Canada)
[3]Geonor T-200B after April 2002, custom adapted Fischer-Porter prior to then, both gauges with Alter shield.

USask established the Peyto Main station, equipped with new instruments (Table 3) and a new setting as a reference station for the PGRB in July 2013 within 20 m from Peyto Main Old (Figure 4c). It measures incoming and outgoing shortwave and longwave radiations, air temperature, humidity, wind speed, precipitation, and snow depth. Figure 5 presents daily averages of these variables for the period from July 2013 to September 2019.



**Table 3: Meteorological measurements and instruments installed at the Peyto Main AWS.**

| Measurements | Units | Instruments | Placements |
|---|---|---|---|
| Air temperature, Ta | °C | Rotronic HC2-S3 Temperature and Humidity Probe | 4.37 m above ground |
| Relative humidity, RH | % | | |
| Wind speed, Ws | m s$^{-1}$ | RM Young 05103AP -10 | 5.23 m above ground |
| Wind direction, Wd | degrees | | |
| Snow temperature, Ts | °C | Omega Type E Thermocouple | 0.2 & 1.5 m above ground |
| Net radiation components: Qsi, Qso, Qli, Qlo | W m$^{-2}$ | Kipp & Zonen CNR4 Net Radiometer | 3.79 m above ground |
| Precipitation, Ppt | mm | TB4 tipping bucket rain gauge | 3.15 m above ground |
| Barometric pressure, P | hPa | Vaisala CS106 | 3 m above ground |
| Snow depth, Sd | m | SR50A Sonic Ranger | 2.95 m above ground |
| Volumetric water content | % | Campbell Scientific CS650 | 0.01 – 0.11 m below ground |
| Electroconductivity | ds m$^{-1}$ | | |
| Soil temperature | °C | | |
| Soil heat flux | W m$^{-2}$ | HFP01 | 0.02 m below ground |

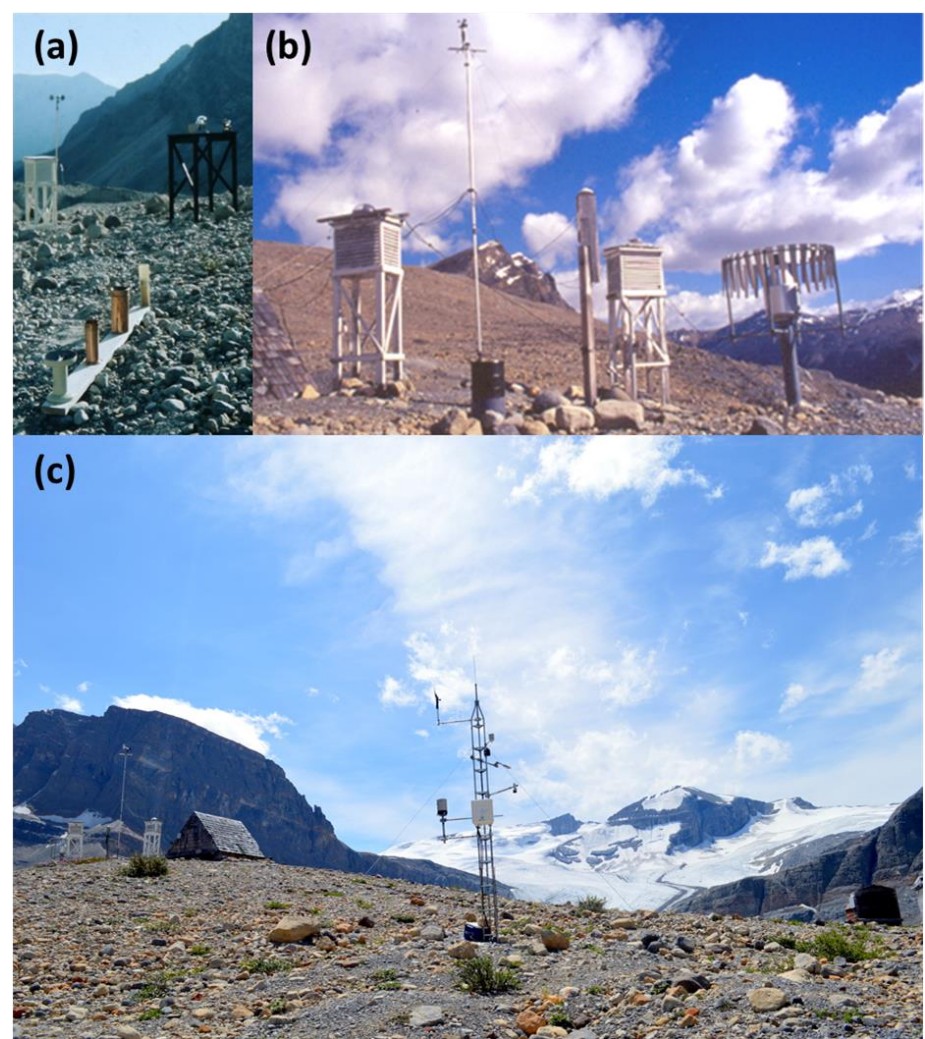

**Figure 4: The base camp stations – (a) Peyto Main IHD, July 1970; (b) Peyto Main Old (July 2009); (c) Peyto Main (Sept 2015) (Peyto Main Old in left background). Photographs by D. Scott Munro (a and b) and May Guan (c).**

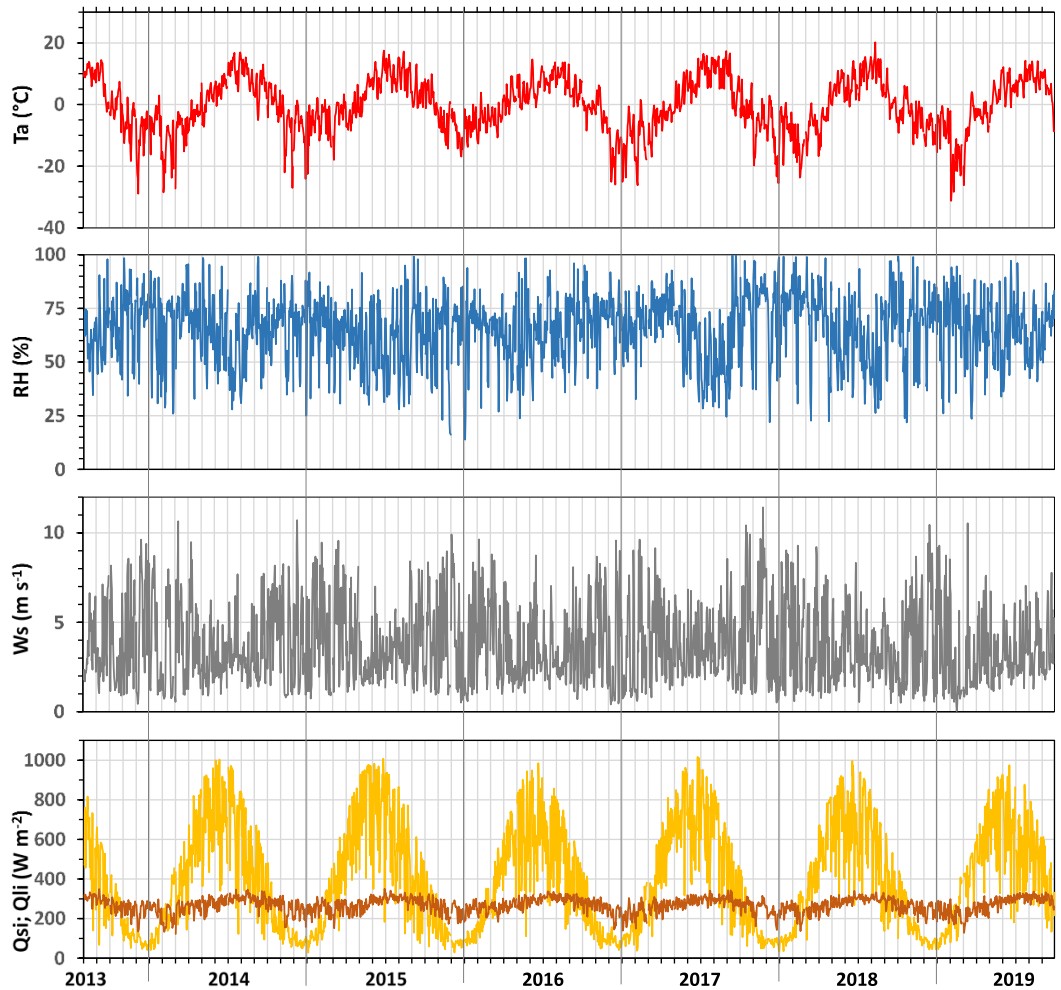

**Figure 5: Peyto Main AWS plots of 24-h mean air temperature (Ta), relative humidity (RH), wind speed (Ws), incoming shortwave radiation (Qsi) and incoming longwave radiation (Qli) – August 2013 to September 2019. Yellow and dark orange in the bottom panel are incoming shortwave and longwave radiation respectively, with 2.75 multiplier applied to shortwave radiation to mimic noon values.**

The nearest AWS outside the basin boundary is operated by USask at the Alpine Club of Canada's Bow Hut (Figure 1), established in October 2012 and continuously monitored since then. Air temperature, humidity, wind speed and snow depth data are available from the station. The Peyto Main AWS and that at Bow Hut were connected to telemetry

10 in 2015, thus enabling them to be monitored remotely. Near real-time data by telemetry, extending back one week can be viewed on the website https://research-groups.usask.ca/hydrology/data.php.

Meteorological data from the Peyto Ice stations (Upper, Middle, and Lower) are not continuous because of difficulties in operating the stations on rapidly ablating glacier ice, but periods of synchronous observational data are available.



The three stations were operational at the same time for brief periods between 2007 and 2013 (Table 1). Peyto Lower Ice has been maintained for a longer period than Middle and Upper Ice, collecting both incoming and outgoing shortwave radiation data until August 2010. Peyto Lower Ice, Peyto Main Old and Peyto Main are currently operational. Peyto Lower Ice was updated with new instruments in October 2015. Station data availability details are

listed in Table 1.

### 3.2   Precipitation

Precipitation at the Peyto Main Old station was measured by a Geonor T-200B, a weighing precipitation gauge with an Alter wind shield, beginning in April 2002, with a CMS tipping bucket (TBRG) rain gauge operating nearby (Figure 4b and Table 2). However, there is reason to doubt the reliability of these records because comparisons with the new

TBRG at the Peyto Main station, 20 metres west of the old station (Figure 4c), show that both the Geonor and the old TBRG recorded significantly less precipitation between June and September (Figure 6), the Geonor catch being approximately 70 % of the new TB catch, that of the old TB much smaller. Also, despite good comparisons with June to September Bow Summit precipitation for 2014 to 2016, just 5.5 km distant (Figure 1), the Geonor persistently underestimates annual precipitation during the six years following 2010 (Figure 7), even though it is 160 m above

Bow Summit.

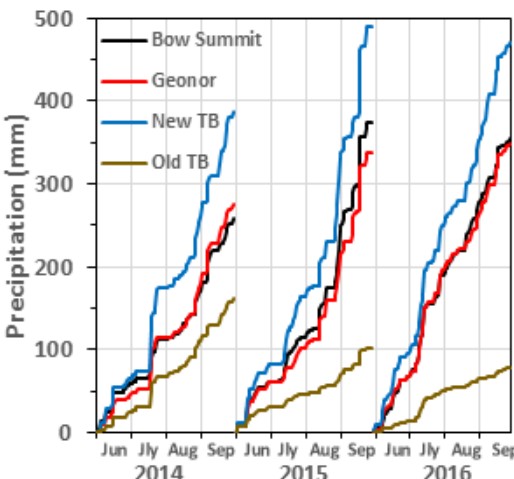

**Figure 6: Cumulative rainfall comparisons at the Peyto Main station and Bow Summit over the summer months.**



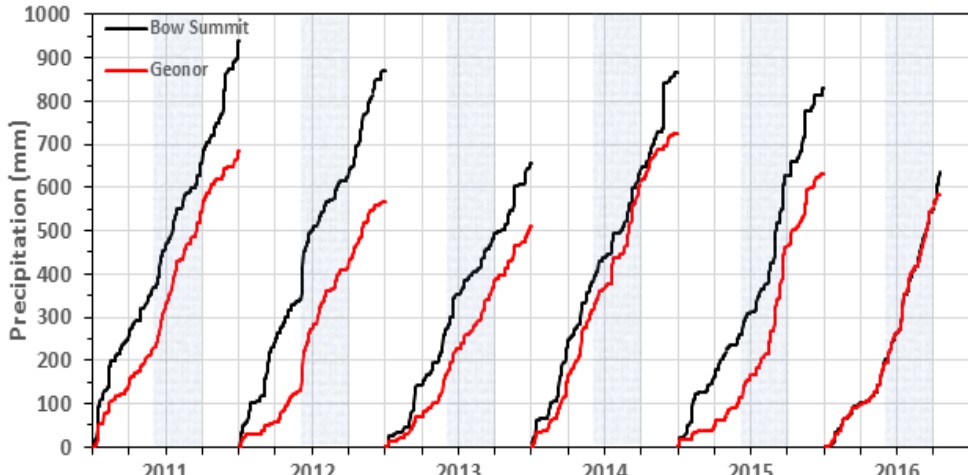

**Figure 7: Annual cumulative precipitation at Bow Summit and Peyto Main, 2011 to September 2016, with highlighting of main summer rainfall months.**

Problems with the old TB date from 2007, when a rapid decline in gauge response was noted (Munro, 2020), but the Geonor gauge response invites further investigation. Therefore, its records were first segregated according to rainfall and snowfall by applying the precipitation phase determination algorithm developed by Harder and Pomeroy (2013). Snowfall was bias-corrected for wind-induced undercatch (Smith, 2007) and rainfall was corrected with a catch efficiency of 0.95 (Pan *et al*. 2016). Bow Summit data were accepted as recorded because the surrounding tall trees

provide sheltering, but do not unload intercepted snow to the single Alter-shielded weighing precipitation gauge at the site (Figure 8), thus making it ideal for precipitation measurements.

Daily precipitation sequences were averaged over seven years, 2010-2016 incl., and seasonally accumulated to compare Peyto Main Geonor and Bow Summit measurements (Figure 9). Observed precipitation accumulations are

similar during the summer months between May and October, with mostly liquid precipitation occurring from June to September. Large differences, however, are found for the adjacent winter snowfall months of January-May and October-December, cumulative winter precipitation recorded at Peyto being significantly less than that at Bow Summit. Therefore, the Peyto precipitation gauge may have been undercatching a large portion of the solid precipitation. It is also possible that the gauge catch correction procedure, originally developed to offset wind induced

undercatch of Canadian Prairie snowfall (Smith, 2007), may require modification for use in a high mountain environment. While the summer precipitation comparisons with the new TB are much closer (Figure 6), the Peyto Main Station is 160 m higher and 5.5 km closer to the continental divide and so would be expected to receive somewhat greater precipitation than a gauge at Bow Summit.





With reservations noted above, the precipitation data recorded at Bow Summit (51.70 N, 116.47 W, Elevation 2080 m, Climate ID: 3050PPF) is considered the most suitable to represent precipitation over the PGRB. Bow Summit data can be downloaded from the Alberta Climate Information Service (ACIS, http://agriculture.alberta.ca/acis/). Quality-controlled hourly temperature and precipitation data are available continuously from 1 November 2008 to the present;

5    continuous daily data are available from 23 March 2006 to the present. The hourly temperature and precipitation data from 1 January 2009 to 31 December 2019 are plotted in Figure 10, earlier data not sufficiently continuous to be included.

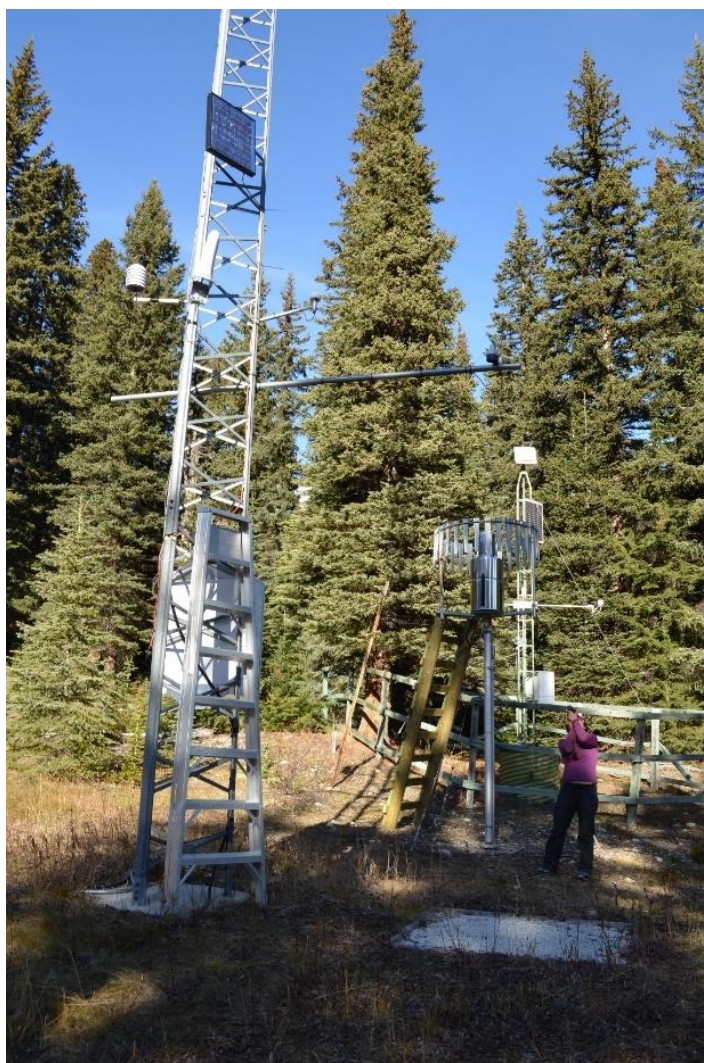

10    **Figure 8: Bow Summit station, 15 October 2015. Photograph by Dhiraj Pradhananga.**

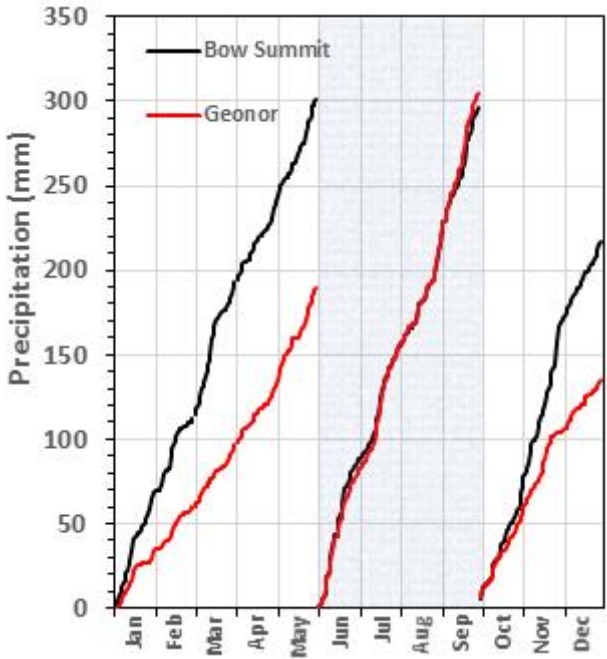

**Figure 9: Seasonal Bow Summit and Peyto Main cumulative precipitation from seven year averages of daily values, 2010 to 2016 incl., with main summer rainfall months highlighted.**

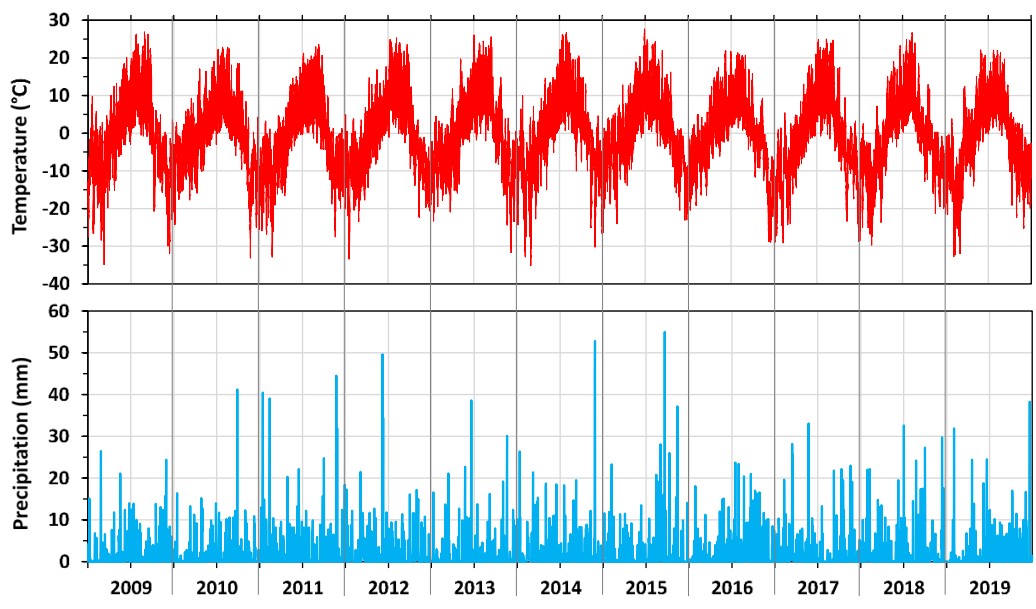

**Figure 10: Hourly temperature and daily precipitation recorded at Bow Summit.**



### 3.3 Data cleaning and gap infilling

Meteorological data recording frequency was changed from hourly to half-hourly in September 2008, with new USask stations recording at 15-minute intervals by 2013 (Table 1). However, quarter and half-hourly data were aggregated

to hourly intervals for archiving, thus corresponding to the AWS recording interval used prior to September 2008. Raw data were thoroughly checked for errors and erroneous data removed. Missing data were filled in by either linear interpolation or linear regression to data from stations within the basin. Linear interpolation was chosen when the data gaps were less than five hours, and linear regressions were applied to longer data gaps. These data cleaning processes were followed in sequence by applying various R functions, along with the CRHMr package (Shook, 2016a) for which

guidance and installation details are available at the GitHub https://github.com/CentreForHydrology/CRHMr. The data processing steps for quality assurance and control are shown in Figure 11.

Despite two data gaps 6-8 months long and five more that span periods of 15-45 days, the Peyto Main Old record is over 91% complete between 1987 and 2012. Gap fill-ins and corrections to key elements, such as air temperature and

solar radiation were done without using the CRHMr package by D. Scott Munro, with flags inserted to aid judgement on data suitability (Munro, 2020). Recent data from Peyto Main Old (4 October 2010 to 31 July 2018) and Peyto Main (17 July 2013 to 1 October 2019) are almost continuous, except for two short gaps in 2013 for Peyto Main Old (13 hours total) and five brief gaps in 2013, 2015, and 2016 for Peyto Main (5.5 hours total) – each a gap of less than four hours. The wind speed data from Peyto Main Old are in error from 17 July 2017 to 8 March 2018. Also, the temperature

and humidity probes at Peyto Main were not functioning properly for longer periods during 2016-2018. The temperature probe at Peyto Main recorded 10 ºC less than that of Peyto Main Old from 22 November 2016 to 8 March 2018 due to a coding error in the datalogger program; the humidity probe was not functioning well from 20 September 2016 to 20 March 2017. These differences were detected by plotting the data and comparing them with data from Peyto Main Old.

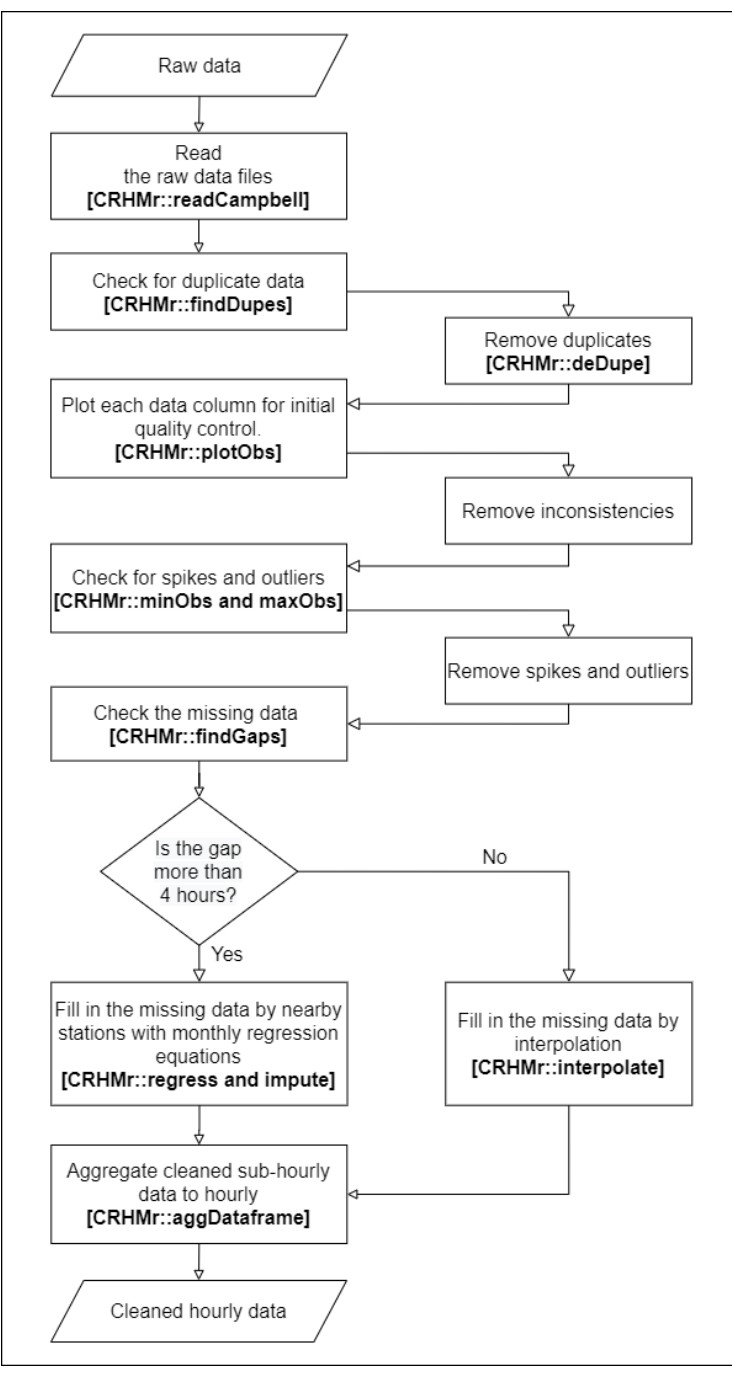

**Figure 11:** Meteorological data cleaning process with corresponding R functions of the CRHMr package stated within brackets



Table 4 shows the regression results and Figure 12 shows the systematic bias in Peyto Main air temperature data before and after a 10 °C correction. The erroneous humidity data were corrected from the Peyto Main Old station data using monthly regressions (Table 5). In addition, Peyto Main station data for all the variables were extended back to 2010 using monthly regressions with data from the Peyto Main Old station.

**Table 4: Regression results for Peyto Main and Peyto Main Old hourly data.**

| Variables | From | To | Slope | Intercept | $R^2$ |
|---|---|---|---|---|---|
| Air temperature | 2013-07-17 | 2018-07-31 | 1.00 | -0.23 | 1.00 |
| Vapour pressure | 2013-07-17 | 2018-07-31 | 1.09 | -0.02 | 0.99 |
| Wind speed | 2013-07-17 | 2018-07-31 | 1.12 | 0.38 | 0.94 |
| Incoming shortwave | 2013-07-17 | 2018-07-31 | 0.96 | 3.39 | 0.97 |
| Incoming longwave | 2013-07-17 | 2018-07-31 | 1.01 | -9.52 | 0.96 |

**Table 5: Monthly regression results for Peyto Main and Peyto Main Old hourly data.**

| Month | Air temperature | | Vapour pressure | Wind speed | Incoming shortwave | Incoming longwave | |
|---|---|---|---|---|---|---|---|
| | Slope | Intercept | Slope | Slope | Slope | Slope | Intercept |
| Jan | 1.00 | -0.26 | 0.99 | 1.19 | 0.91 | 1.00 | -6.12 |
| Feb | 0.99 | -0.24 | 1.00 | 1.18 | 0.94 | 1.00 | -6.92 |
| Mar | 0.99 | -0.29 | 1.01 | 1.17 | 0.95 | 1.01 | -8.97 |
| Apr | 1.00 | -0.24 | 1.03 | 1.17 | 0.97 | 0.99 | -6.48 |
| May | 1.00 | -0.32 | 1.06 | 1.15 | 0.98 | 1.04 | -20.52 |
| Jun | 1.01 | -0.28 | 1.07 | 1.18 | 0.98 | 1.04 | -19.02 |
| Jul | 1.00 | -0.14 | 1.08 | 1.17 | 0.96 | 1.04 | -19.69 |
| Aug | 1.00 | -0.24 | 1.07 | 1.21 | 0.95 | 1.03 | -16.51 |
| Sep | 1.01 | -0.34 | 1.05 | 1.22 | 0.96 | 1.04 | -17.11 |
| Oct | 1.01 | -0.23 | 1.04 | 1.22 | 0.95 | 1.05 | -18.75 |
| Nov | 1.00 | -0.24 | 0.98 | 1.20 | 0.94 | 1.05 | -18.17 |
| Dec | 1.00 | -0.22 | 0.99 | 1.20 | 0.91 | 1.01 | -9.58 |

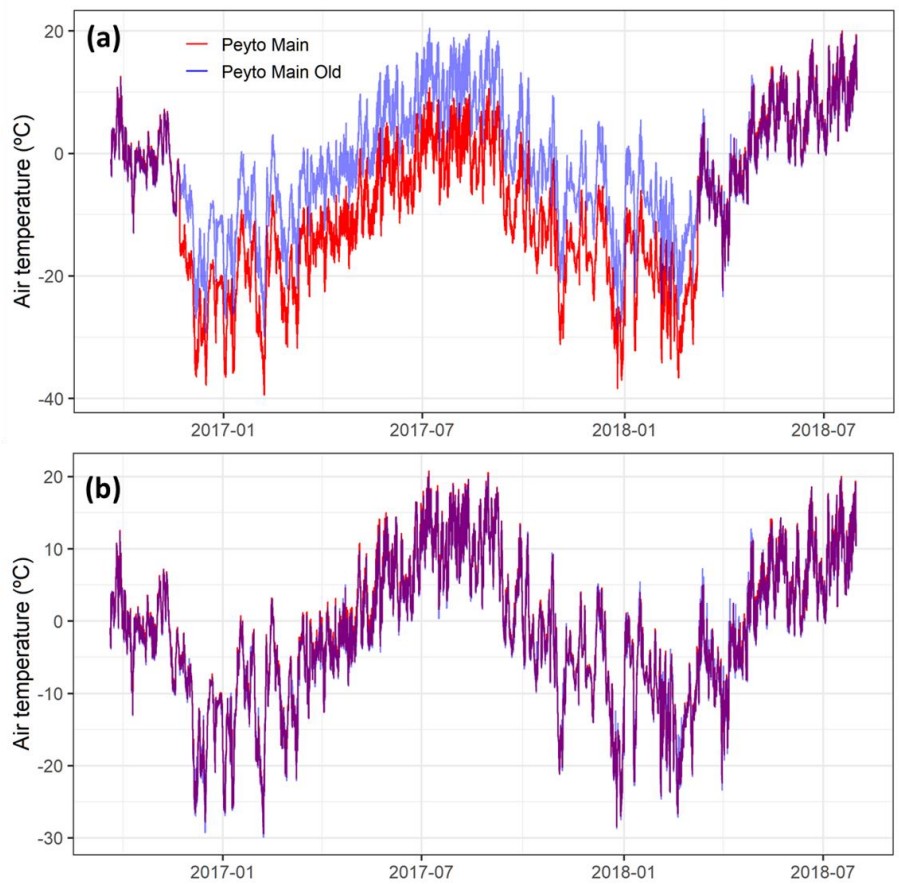

**Figure 12: Air temperature recorded at the Peyto Main and Main Old stations: (a) before bias correction to Peyto Main, (b) after bias correction. Overlapping values appear in purple.**

5     **3.4   Reanalysis forcing data**

Bias-corrected reanalysis data are also included as model forcing data for running glacio-hydrological models over long periods. Four gridded reanalysis products were bias corrected, using *in-situ* observations at the PGRB:

1.  CFSR, the Climate Forecast System Reanalysis product (Saha *et al.*, 2010).

2.  ERA-Interim, the European Centre for Medium-Range Weather Forecasts Interim reanalysis product (Dee et
10      al., 2011);

3.  NARR, the North American Regional Reanalysis product (Mesinger *et al.*, 2006); and

4.  WFDEI, the Water and Global Change (WATCH) Forcing Data ERA-Interim reanalysis product (Weedon et al., 2011).



These products are available at different spatial and temporal resolutions for different time periods. CFSR, ERA-Interim, and WFDEI are global datasets, whereas NARR covers only North America. ERA-Interim is available from January 1979 to 2018, with original resolution of 0.7º at the Equator (Dee et al., 2011). WFDEI (Weedon et al., 2011) is available at a spatial resolution of 0.5º x 0.5º from 1979 to 2016. NARR (Mesinger et al., 2006) is available at 3-

5   hourly temporal and 32 km spatial resolutions from January 1979 to January 2017.  CFSR, developed by the National Center for Environmental Prediction and the National Center for Atmospheric Research (NCEP-NCAR), is available hourly, at a horizontal resolution of $0.5° \times 0.5°$ from 1979 to 2009 (Saha et al., 2010). A comparison of three reanalysis products showed ERA-Interim to be better than NARR and WFDEI for air temperature, vapour pressure, shortwave irradiance, longwave irradiance and precipitation, while WFDEI was best for wind speed (Pradhananga, 2020).

All gridded reanalysis data were first extracted for the Peyto Main station coordinates. ERA-Interim, WFDEI, and NARR data were interpolated to hourly time periods. The R-package, Reanalysis (Shook, 2016b) was used for extracting and interpolating ERA-Interim, WFDEI, and NARR datasets. Air temperature, vapour pressure, wind speed, precipitation, incoming longwave and incoming shortwave radiation data were interpolated linearly from 3 or

15   6 hour to hourly time intervals. Total precipitation (3 or 6 hours) was distributed evenly to hourly time intervals. MATLAB (MATrix LABoratory) codes (Krogh *et al.*, 2015) were used to extract CFSR values, which were already at hourly time intervals. The hourly data were bias-corrected to the *in-situ* observations at the main station, using a quantile mapping technique, with parameters calibrated for each month from corresponding data periods using the qmap package in R (Gudmundsson, 2016). ERA-Interim data from January 1979 to August 2019 are presented in

20   Figure 13.

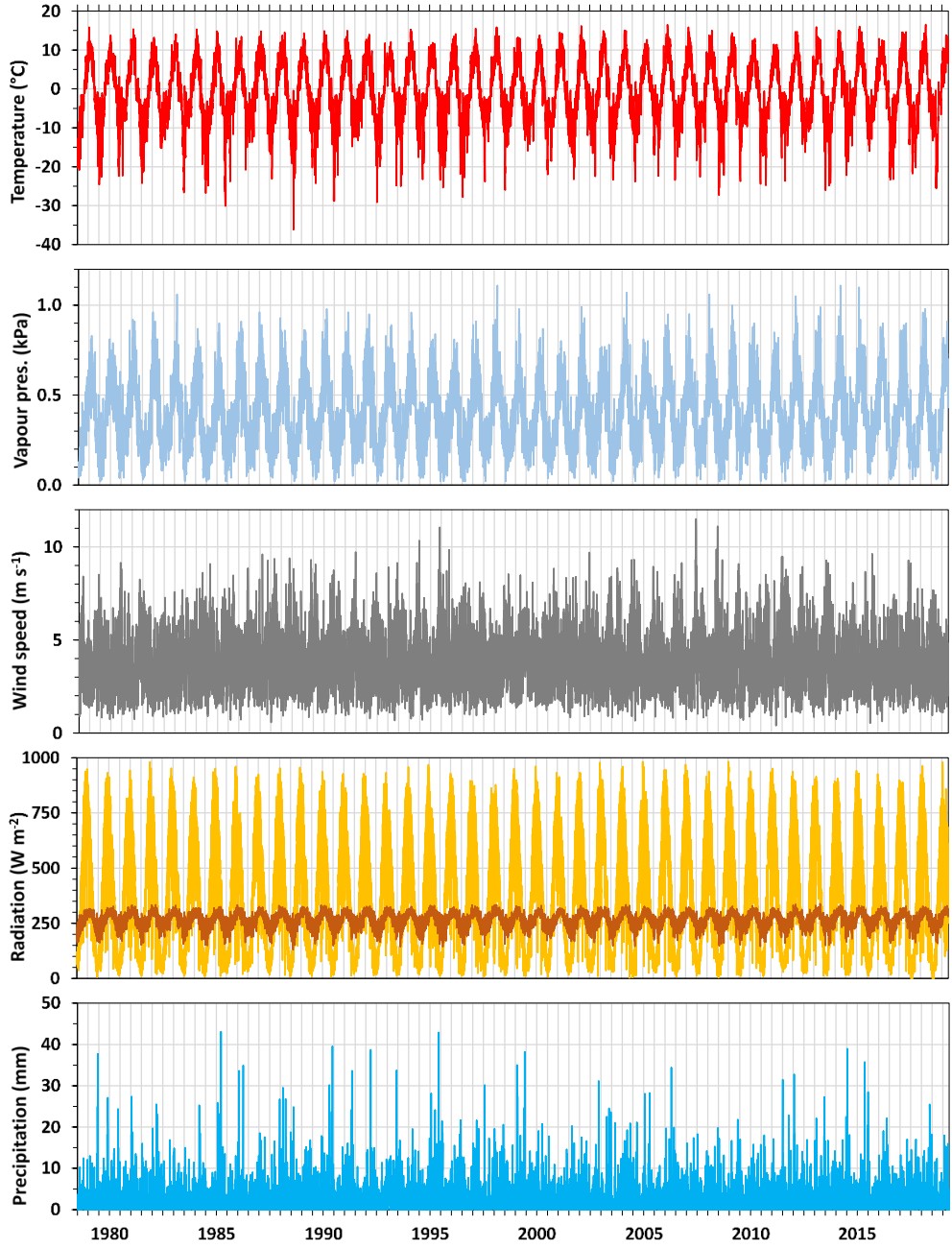

**Figure 13: Bias corrected meteorological forcing data from ERA-Interim. Daily precipitation is the 24-hour total, and the other data are plotted as 24-h means. Yellow and dark orange in the radiation panel are incoming shortwave and longwave radiation respectively, with 2.75 multiplier applied to shortwave radiation to mimic noon values.**





### 3.5 Hydrological data – historical and present

Historical observed daily outflows from the glacier at Peyto Creek are available for 1967 to 1977, from the Water Survey of Canada (WSC, https://wateroffice.ec.gc.ca/search/real_time_e.html). They are also available at 15-minute intervals from 1970 to 1977 by accessing the Peyto Glacier runoff archive housed at the University of Waterloo
(Munro, 2011a). The gauge station (ID 05DA008) was established in 1966 for the IHD program and maintained by the WSC. It consisted of a float-activated continuous stage recorder (Table 6) mounted on a stand pipe ~500 m from the glacier tongue (Figure 14).

Historical discharge measurements at Peyto Creek are problematic due to unstable cross-sections, occasional flash
floods and lack of direct discharge measurements during high flows. Goodison (1972) reported that the discharge records from 1967 are not reliable, and the stage gauge was washed out during a flood in August 1967. As reported by Ommanney (1987), heavy precipitation and a resulting landslide in July, 1983 triggered two floods. The instantaneous discharge during the first flood was estimated to be in the range of 200 to 300 m$^3$ s$^{-1}$ (Johnson and Power, 1985), and an estimated 6000 m$^3$ of debris, approximately 3 m thick, was deposited in the valley near the
gauging site. A similar event in September 2010 deposited a thick debris cover over the original gauge area, thus changing the trail into the glacier.

A new hydrometric station to resume flow measurements for Peyto Creek was installed at the outflow of Lake Munro on the bedrock near the glacier snout in 2013 by USask (Figure 14b). It is 1.5 km upstream from the old gauging site
and so redefines the gauged basin to a smaller area (Figure 1). The new station is equipped with a Campbell Scientific Sonic Ranger (SR50A) to monitor the water stage. This gauge record is temperature corrected using air temperature measured below the SR50A. In the summer of 2018, an automated salt dilution system and a stage level logger were installed approximately 100 m downstream of the SR50A. Between 14 May 2018, and 10 September 2018, 43 streamflow discharge measurements were performed with automated and manual salt dilutions. One manual
streamflow measurement was conducted with an FT2 handheld Acoustic Doppler Velocimeter (ADV) on 1 August 2018. Early season measurements were taken when the stream upstream of the survey site was still snow-covered. Sudden drops in the stage were observed during that period, likely due to temporary ice jamming. Therefore, two rating curves were developed for the 2018 season (Sentlinger et al., 2019), one for the early season when the stream was still snow-covered, and the other for the melt season, when the stream was snow-free. For the 2018 season, the
shift between early and late rating curves occurred on 12 June. When using this rating curve to obtain streamflow for 2013-2018, only the late-season curve is used, as the SR50A site became snow-covered in winter and measurements were only available after snowmelt exposed the stream in spring. The daily mean basin runoff (streamflow discharge per unit area of the basin) averaged over the historical 11-year period (1967-1977) and the present 5-year period (2013-2018) are presented in Figure 15.





**Table 6: Hydrometric station information.**

| Hydrometric station | Geographical coordinates | Drainage area | Elevation above sea level | Stage recording instrument and rating curve method | Discharge data period of record |
|---|---|---|---|---|---|
| Old gauge: Peyto Creek at Peyto Glacier (05DA008) | 51.69361 N 116.53556 W | 23.6 km$^2$ | 1951 m | Stevens A-35 water-level recorder; rating curve data from current meter for low flows, salt dilution or Rhodamine dye injection for high flows (Goodison, 1972) | 1967-1977 (June – Sept) |
| New gauge: Lake Munro outlet | 51.68111 N 116.54472 W | 18.3 km$^2$ | 2150 m | Campbell Scientific SR50 ranger; rating curve data from salt dilution method | 2013-2018 (June – Sept) |

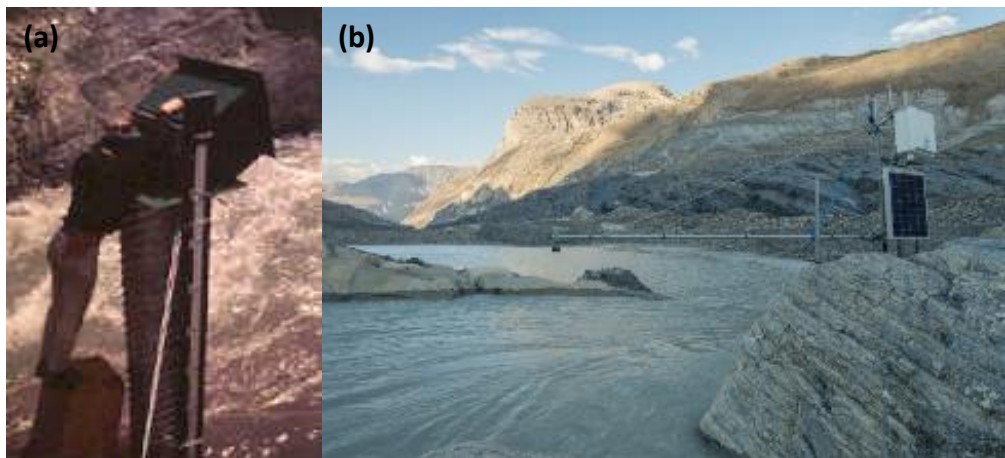

**Figure 14: Gauge sites: (a) old IHD hydrometric gauge on Peyto Creek, August 1970 and (b) new hydrometric station at the Lake Munro outlet. Photographs by D. Scott Munro (a) and Angus Duncan (b).**

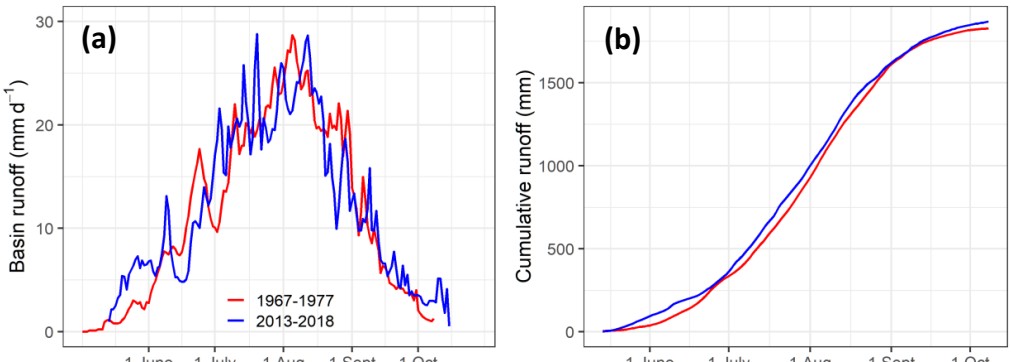

**Figure 15: Runoff data: (a) Daily basin streamflow, expressed as a depth of runoff per day, averaged over the historical 1967-77 and recent 2013-18 periods; (b) Cumulative annual depth of runoff averaged over the same periods.**

### 3.6 Glaciological data

Glaciological mass balance measurements, using ablation stakes and snow pits, have been taken semi-annually since 1965, when the IHD program began, the scheme for Peyto Glacier was first described by Østrem (1966). Mass balance data for 11 elevation bands, 100 m in width, are reported in several publications (Demuth et al., 2009; Demuth and

Keller, 2006; Dyurgerov, 2002; Ommanney, 1987; Young, 1981; Young and Stanley, 1976). Recent mass balance data are available from the WGMS (http://www.wgms.ch). The WGMS (2019) has also compiled datasets from 1966 to 2017 that are plotted in Figure 16 (1991-1992 mass balance year missing). Specific winter and summer mass balance data for 11 elevation bands covering an elevation range from 2100 to 2703 m are also available for the period 2003-2018 that are not included in this study. The winter, summer, and annual point balances have been calculated for the

middle of each elevation band, from 2150 to 2650 m above sea level, using a local polynomial regression technique.

The dataset also includes frontal variation, equilibrium line altitude (ELA), accumulation area ratio (AAR), glacier mass balance (winter, summer, annual) and repeat photographs (WGMS, 2019). Radio detection and ranging (radar) measurements of ice thickness for Peyto Glacier in the 1980s were reported by Holdsworth *et al.* (2006). Ground-

penetrating radar surveys of ice thickness across the glacier tongue in 2008-2010 were reported by Kehrl et al. (2014) in their study of volume loss from the lower Peyto Glacier area between 1966 and 2010.




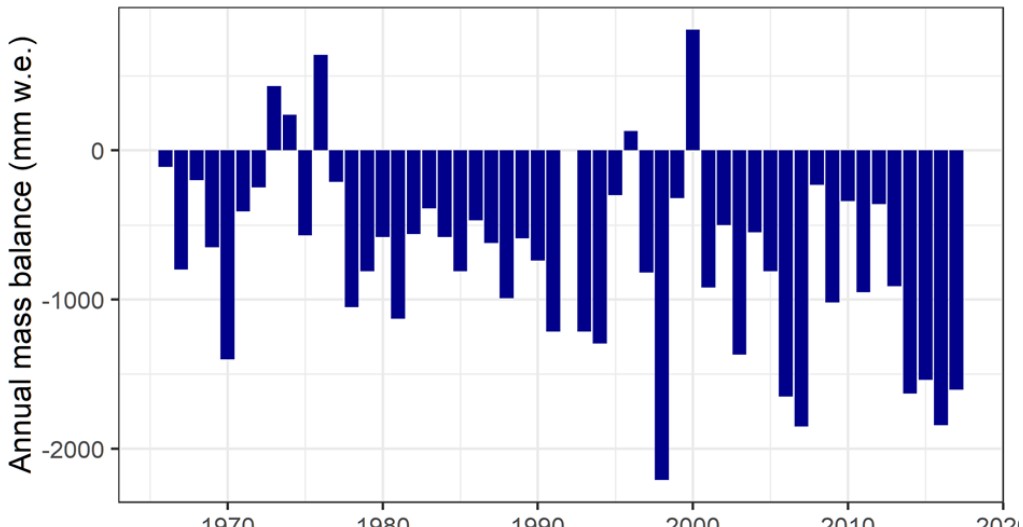

**Figure 16: Net annual mass balance data for Peyto Glacier. Data source: WGMS (2019).**

It should be noted that in several instances the data sets feature variations in temporal subsets of the data. An example
is the WGMS record which, for a portion of the record, utilizes data from the Dyurgerov (2002) synthesis rather than
Environment Canada National Hydrology Research Institute observations compiled by Ommanney (1987). Moreover,
all data sets present a mix of reference-surface mass balance data, with hypsometry held constant, and conventional
mass balance data, where hypsometric changes are reflected in mass balance accounting (Cogley et al., 2011).

**3.7   Geospatial data**

**3.7.1   Digital elevation models (DEMs)**

Repeat DEMs can be used to quantify surface height changes through time, which are then converted to mass change.
Photogrammetric techniques have been to construct a high-quality DEM from 1966, and airborne Light Detection and
Ranging (LiDAR) surveys were used to collect DEMs for 2006 (Demuth and Hopkinson, 2013) and 2017 (Pelto et

al., 2019; Table 8). The 2006 DEM was obtained from the Geological Survey of Canada and the Canadian Consortium
for LiDAR Environmental Applications Research. DEMs from 1966 and 2006 were co-registered to the 2017 DEM
based on the algorithm proposed by (Nuth and Kääb, 2011) using an automated, open-source tool developed by
Amaury Dehecq (https://github.com/GeoUtils). DEM sources, preparation, and co-registration are described below
and presented in Table 8.



### 3.7.1.1 Generation of photogrammetric DEMs

Digital copies of diapositives from the year 1966, photogrammetrically scanned at a resolution of 14 μm, were obtained from the Canadian National Air Photo Library. The photographs were taken near the end of the ablation season (Table 7). However, there was extensive fresh snow cover in the images that resulted in poor contrast in the accumulation region of the glacier.

The DEM was generated using the Agisoft Metashape Professional (AMP) Edition, Version 1.5. All photos were assigned to the same camera group based on the focal length, pixel size and fiducial coordinates available from the camera calibration report. Then the photos were aligned by AMP and a sparse point cloud model was produced in which camera positions and orientations are indicated. To optimize the camera positions and orientation data, some reference points (GCPs) were identified from the stable terrain surrounding the glacier, over a range of elevations. The GCP file was imported to AMP, and corresponding locations were marked on each of the photos. Finally, based on the estimated camera positions, AMP calculated depth information and a dense point cloud was generated. A DEM and an ortho image were produced from the dense point cloud.

Most of the accumulation zone of the glacier is missing from the dense point cloud because fresh snow cover resulted in poor contrast in this region. The interpolation feature available in AMP was not enabled whilst generating the DEMs, as it does not generate very accurate elevations. The spatial resolution of the DEM was chosen to be 10 m.

**Table 7: Aerial photographs used.**

| Year | Date | Data Source | ID | No. of Photos | Scale | Accumulation Area Contrast | No. of GCPs |
|------|------|-------------|-----|---------------|-------|----------------------------|-------------|
| 1966 | 20 Aug | Federal AP | A18434 | 5 | 1:40000 | Poor | 18 |

### 3.7.1.2 Generation of LiDAR DEM

Light Detection and Ranging (LiDAR) uses a laser pulse to calculate the distance of the target from the sensor. An airborne laser survey was conducted using a Riegl Q-780 full waveform scanner and Applanix POS AV Global Navigation Satellite System (GNSS) Inertial Measurement Unit (IMU). The laser survey trajectory data was processed using PosPac Mobile Mapping Suite (Applanix) resulting in horizontal and vertical positional accuracy typically better than ±15 cm. RiPROCESS was used to post-process the point clouds and export to a LAS (LiDAR data exchange file) format, a binary file to store LiDAR data. LASTools, available from https://rapidlasso.com/lastools/, was used to process the point cloud and generate the DEM (Pelto et al., 2019).

### 3.7.1.3 DEM co-registration

It is important to align the multi-temporal DEMs relative to one another so that the same point on the ground is represented at the same location in each DEM, thus enabling glacier elevation change to be measured as accurately as possible (e.g., Figure 17). The 2017 LiDAR DEM was taken as the master DEM and all other DEMs (Table 8) were

co-registered with respect to this DEM following the Nuth and Kääb (2011) method. The 1966 ortho image was used to mask out all the unstable areas such as glaciers, fresh snow, or water bodies. All the pixels outside this mask were classified as stable terrain and thus used for co-registration, using the github repository at https://github.com/GeoUtils to perform the task, the co-registration statistics listed in Table 9.

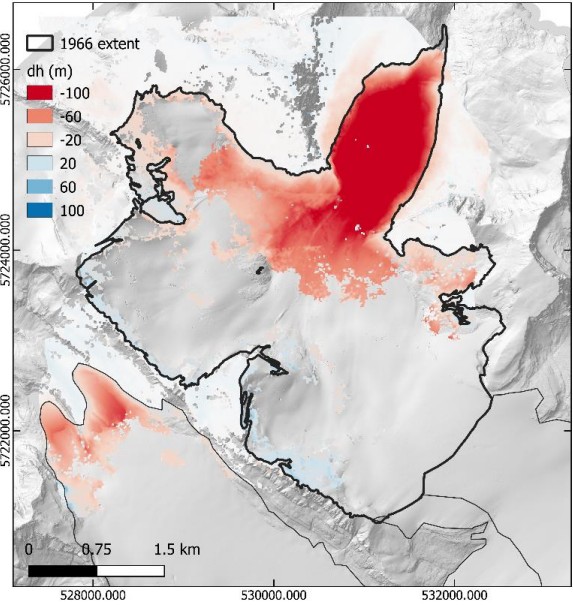

**Figure 17: Elevation change over Peyto Glacier, 1966-2017, inside the IHD glacier boundary.**

**Table 8: DEMs used for co-registration.**

| Year | Resolution | Source and method |
|------|-----------|-------------------|
| 1966 | 10 m | This DEM was prepared from digital copies of diapositives, photogrammetrically scanned at 14 μm resolution, obtained from the Canadian National Air Photo Library. A 10 m resolution DEM was generated using AMP Edition, Version 1.5. |
| 2006 | 10 m | This DEM was prepared from LiDAR surveys taken in August 2006 (Demuth and Hopkinson, 2013). The DEM did not cover the whole area of the PGRB, so the northeast corner of the basin was mosaiced with a 2014 DEM data to fill in the missing part. |
| 2017 | 1 m | This DEM was prepared from LiDAR surveys taken on 17th September 2017 and is available in the archive of the University of Northern British Columbia (UNBC). |

**Table 9: Stable terrain statistics after co-registration.**

| Co-registered DEM (year) | Master DEM (year) | Median (m) | Normalized Median Absolute Deviation (m) |
|--------------------------|-------------------|------------|-------------------------------------------|
| 1966 | 2017 | -0.25 | 8.91 |
| 2006 | | -0.07 | 1.00 |



### 3.7.2 Landcover data

Landcover changes in the PGRB were compiled from remotely sensed imagery. ESRI ArcGIS, Agisoft and R were used to work with the time series of Landsat images and digital elevation models (DEM), using various tools and functions available in the software modules. Google Earth Engine (GEE) was also used for the spatial and temporal analysis of annual landcover mapping from Landsat images. Landcover maps from the satellite images were prepared by classification in accordance with albedo, the normalized-difference snow index (NDSI), and the normalized-difference water index (NDWI). As datasets extracted from different sources have different projection systems, they were re-projected to NAD 1983 UTM Zone 11 (EPSG: 26911).

#### 3.7.2.1 *Basin delineation and landcover classification*

The PGRB drainage basin was delineated from the 1966 DEM. Google Earth Engine was used for the landcover classification of Landsat images of each year, from the 1980s to the present. Landcover information was extracted from Landsat 5 and Landsat 8 top-of-atmosphere (TOA) reflectance images. Landsat 5 images were used for the years 1984 to 2011, Landsat 8 images from 2013 to 2016. The Landsat satellite images are freely available and accessible through GEE at 30 m spatial resolution and 16-day temporal resolution. Two criteria governed image acquisition: (a) an image date between 15th July and 15th September; (b) minimal or no cloud cover inside the PGRB boundary. Landsat images used to create landcover classification of the PGRB appear in Table 10. Landsat 5 images were from the Thematic Mapper (TM) sensor, Landsat 8 images from the Operational Land Imager (OLI). Images for the years 1992, 1995, 1999 and 2012 are missing due to failure to meet the criteria. A landcover map for 1966 was created from the topographic map of Sedgwick and Henoch (1975).

Four landcover classes were identified: (1) firn/snow (accumulation area), (2) ice (ablation area), (3) bare (non-glacierized area) and (4) water body. Snow and non-snow covered areas of bare landcover were differentiated by the NDSI (Hall *et al.*, 2002) and NDWI (Gao, 1996; McFeeters, 1996). Snow and firn areas within firn/snow landcover were classified by their albedo (Liang, 2000; Smith, 2010). The NDSI, NDWI and albedo for the images were obtained from the Raster Calculator on the GEE platform. Accordingly, landcover classification proceeded as follows:

1. **Bare**: all snow-free non-glacierized areas identified by the NDSI
2. **Firn/Snow**: glacierized areas with albedo greater than 0.4 and NDWI lower than 0.4
3. **Ice**: glacierized areas with albedo lower than 0.4 and NDWI lower than 0.4
4. **Waterbody**: Areas with NDWI greater than 0.4

After GEE export to Google Drive the images were downloaded from the drive and converted to a shape file using the Raster to Polygon tool in ArcMap.





**Table 10: Landsat images for generating landcover maps of the PGRB**

| Landsat | Year | Date | Image ID |
|---------|------|------|----------|
| Landsat 5 | 1984 | August 15 | LT50430241984228PAC00 |
| | 1985 | August 02 | LT50430241985214PAC02 |
| | 1986 | August 28 | LT50440241986240XXX01 |
| | 1987 | August 08 | LT50430241987220XXX02 |
| | 1988 | September 02 | LT50440241988246XXX01 |
| | 1989 | August 13 | LT50430241989225XXX02 |
| | 1990 | August 07 | LT50440241990219PAC00 |
| | 1991 | September 04 | LT50430241991247XXX02 |
| | 1993 | August 15 | LT50440241993227PAC03 |
| | 1994 | August 11 | LT50430241994223PAC02 |
| | 1996 | August 23 | LT50440241996236PAC00 |
| | 1997 | August 03 | LT50430241997215PAC03 |
| | 1998 | August 29 | LT50440241998241PAC03 |
| | 2000 | August 18 | LT50440242000231XXX01 |
| | 2001 | August 14 | LT50430242001226LGS02 |
| | 2002 | August 24 | LT50440242002236LGS01 |
| | 2003 | August 20 | LT50430242003232PAC02 |
| | 2004 | August 13 | LT50440242004226EDC00 |
| | 2005 | August 09 | LT50430242005221PAC01 |
| | 2006 | August 28 | LT50430242006240PAC01 |
| | 2007 | August 15 | LT50430242007227PAC01 |
| | 2008 | August 17 | LT50430242008230PAC02 |
| | 2009 | August 27 | LT50440242009239PAC01 |
| | 2010 | August 14 | LT50440242010226PAC01 |
| | 2011 | August 26 | LT50430242011238PAC01 |
| Landsat 8 | 2013 | August 22 | LC80440242013234LGN00 |
| | 2014 | August 18 | LC80430242014230LGN00 |
| | 2015 | August 12 | LC80440242015224LGN00 |
| | 2016 | August 30 | LC80440242016243LGN00 |

## 4 Data availability

All datasets described and presented in this paper can be openly accessed from the Federated Research Data Repository

5     at https://doi.org/10.20383/101.0259 (Pradhananga et al., 2020). All meteorological and hydrological data are reported in mountain standard time (MST) that is 7 h behind Greenwich mean time (GMT − 7). Meteorological, both in-situ and bias-corrected reanalysis products, are time series in tab-delimited .obs text files. They are readable directly by CRHMr functions and any Cold Region Hydrological Model, CRHM (https://research-



groups.usask.ca/hydrology/modelling/crhm.php). Glacier mass balance and streamflow datasets are in .csv files. Geospatial data, co-registered DEM and landcover shapefiles are provided in NAD 1983 UTM Zone 11 projection.

## 5   Summary

This paper describes the hydrometeorological, glaciological and geospatial data collected at the Peyto Glacier Research Basin over the past five decades from its foundation by the Government of Canada as part of its contribution to the UNESCO International Hydrological Decade.  The research basin now forms part of the Canadian Rockies Hydrological Observatory and so has been extensively re-instrumented and subject to intensive scientific study in the last decade. The meteorological data are from six AWS sites, three on the glacier and three near the glacier. These stations are listed as CryoNet stations of the WMO GCW. Near real-time data from Peyto Main and Bow Hut are publicly accessible through telemetry at

https://research-groups.usask.ca/hydrology/data.php#CanadianRockiesHydrologicalObservatory.

Several examples of data cleaning approaches are presented. The Peyto Main station was operational during the summer months of the IHD and re-established as an AWS in 1987. New instruments and dataloggers were added in 2012-2013 by the Centre for Hydrology, University of Saskatchewan. The meteorological data include hourly air temperature, humidity, wind speed, incoming shortwave and longwave radiation, and precipitation. These data are available for a period longer than two decades from the Peyto Main station, and for longer than one decade from the on-ice stations. Bias-corrected ERA-Interim (European Centre for Medium-Range Weather Forecasts Interim reanalysis), WFDEI (Water and Global Change Forcing Data ERA-Interim), NARR (North American Regional Reanalysis), and CFSR (Climate Forecast System Reanalysis) data are also included for running hydrological models over longer periods.

Glaciological mass balance data are collected semi-annually by Natural Resources Canada and partners and published by the WGMS and updated annually. Details of these data have been described in several publications. Specific mass balance data at different elevation zones, available from 2007 to 2019, are included in this paper. On-ice station data include glacier surface elevation change due to ablation and accumulation, as measured by sonic rangers at three ice stations. The three ice stations, each in a different elevation zone, have been operational for various time periods, the first starting in 1995, with long gaps in the records becoming less frequent over time, especially after 2007. Geospatial data include information on basin boundary, drainage area, landcover (including snow, firn and ice on the glacier), and locations of hydrometric sites. Both historical and contemporary discharge data are included. The flow data and hourly surface elevation change data in different elevation zones can be useful for model validation. The long-term mass balance data are a valuable research asset for model development, analysis of climate change, and study of climate impacts on glacier mass balance and hydrology.  This comprehensive, exceptionally long database is a testament to the dogged perseverance of scientists working for various entities with support from various research funding schemes who kept their eyes on the science and so have produced a rare half-century detailed documentation of the impacts of climate change on the cryosphere in a high mountain environment.



**Author contribution**

DP cleaned, organized, and corrected the data and wrote the first draft of the manuscript. JWP and DSM designed and instrumented the research basin. All the authors collected data and contributed to the paper writing.

**Competing interests.**

The authors declare that they have no conflict of interest.

**Disclaimer**

Any reference to specific equipment types or manufacturers is for informational purposes and does not represent a product endorsement.

**Acknowledgements**

This paper and sustained observations at the PGRB were made possible by funding from the Global Water Futures programme supported by the Canada First Research Excellence Fund; the Changing Cold Regions Network and Discovery Grants supported by the Natural Sciences and Engineering Research Council of Canada; the Canada Research Chairs Programme; the Canada Excellence Research Chair in Water Security; the IP3 and WC2N Networks supported by the Canadian Foundation for Climate and Atmospheric Sciences; the Canada Foundation for Innovation,

Natural Resources Canada, and Environment Canada. The 2017 airborne LiDAR survey of Peyto Glacier was supported by the Columbia Basin Trust and BC Hydro. More than a half-century of intense field observations on a remote high mountain glacier in the Canadian Rockies involve extraordinary dedication, perseverance, foresight and bloody-mindedness along with the physical fortitude to take scientific measurements in inclement weather.  This paper is dedicated to the many brave scientists who have taken observations on Peyto Glacier and to Dr. Gordon Young,

who has not only done all of that but continues to encourage scientific examination of the glacier and of the dynamic interface of the cryosphere and the hydrosphere.



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
