# Peer review of "Hydrometeorological, glaciological and geospatial research data from the Peyto Glacier Research Basin in the Canadian Rockies"

_Earth System Science Data, 2020_

## Referee Comment (RC1) · Anonymous Referee #1 · 6 Nov 2020

The authors present hydrometeorological and glaciological observations of the Peyto Glacier Research Basin in Canada, including high resolution DEMs, long term meteorological data, precipitation, outflow from the glacier and bias corrected reanalysis information. In addition, the authors provide an interesting description of the historical monitoring efforts of the basin. In this respect, I found the dataset a good contribution to the already available information over mountainous regions that could asses the basis of future glaciological studies in the area. In general, I find the dataset very valuable and self-explanatory, however the description of the dataset in the manuscript is sometimes a bit confusing. Some paragraphs mix the methods to process and validate the data, and the data available from other sources, with the description of the dataset

itself.

My mayor concern about the dataset is the chosen platform to share the data. It does not allow simultaneous downloads, unless the user registers on it. This is highly annoying as the user have to download and install specific software from the platform. Such software comes with third party libraries that Linux users have to install manually. It seems like such libraries are not totally supported by all systems, or are not totally supported in an "user-friendly" way. Thus, it is very likely that the linux users have to use specific command line tools, just to download few Mbs of data. Furthermore, we found the platform surprisingly slow in comparison with other repositories. The authors should consider migrate the dataset to other platform, or at least submit a single file compressed version of the whole dataset that will allow future users to download the complete dataset avoiding the use of the platform GUI application and registration.

Some specific comments bellow:

Table1: Maybe I misunderstood something, but I can not find in the database the same variables reported in the Table 1. For instance, Peyto Main should include [Ta, RH, Ws, Wd, Ts, Qsi, Qso, Qli, Qlo, Ppt, P, Sd], but the file in the dataset includes just [Ta, RH, Ws, Qsi, Qli]. Why not to include all the variables?.

11p/ Lines 5 to 10. It is unclear if the data provided is the raw information or the corrected one. If it is the raw data please highlight, if not, the raw data should be included. I miss a brief description of the followed methodology for the bias correction of the solid precipitation for wind induced undercatch.

Figure 11. Is the gap filling procedure applied when the gap is bigger than 4 hours or lower than 5 (form the text)? It will be interesting to flag the filled timesteps.

It is surprising that the authors have chosen the deprecated ERA-Iterim reanalisys instead of ERA5. Is there any reason for that? I am not familiar with WFDEI, but is it not just a bias corrected version of the Era-Iterim with a spatial interpolation to

improve the resolution? If the idea was to perform a bias correction, Why not to use just ERA5-Land reaching a much higher resolution?.

19p/ Lines 12-20. As is presented as forcing data, it would be interesting for users to know the elevation and coordinates of the original reanalysis cells. If after the bias correction the elevation of the reanalysis cells should be considered the same as for main station, please highlight. Why not to use other stations too?. It will be interesting to read few comments about the limitations of bias corrected reanalysis specifically in terms of resolution compared with the distributed in situ observations. In addition, the metadata of the dataset highlights that the bias correction was performed using different stations (Metadata: "Bias corrected to Peyto Main for teauQsiQli, and to BowSummit for p"). It should be specified and justified in the text.

3.6 Glaciological data. Not all this data sources are available in the link provided by the authors (e.g. repeat photography). Please clarify when you are summarizing the available information, and when you are describing the new dataset, specifically in the second paragraph.

Please, use same coordinate reference systems for all the geospatial data. (e.g. BasinBoundary).

25p/Line 10. Could you provide a description of these "stable terrain"?.

26p./Line 3 to 4. the authors have probably used the tool GeoUtils, not the repository it self. I find the whole sentence confusing.

27p/Line 20. Again, this 1966 Land cover map is not available in the link provided by the authors.

---

## Referee Comment (RC2) · Anonymous Referee #2 · 14 Jan 2021

This paper presents a valuable collection of long term glaciological, hydrological and meteorological observations in a well-studied glacier-fed catchment. Although most of the datasets were already used elsewhere the authors argue that these datasets have "never been assembled in a single description until now". I am not familiar with the study region so I cannot verify this statement. However, it may be useful to emphasize which dataset was already published and which one was not (maybe in a table). The paper reads well in general, but some sections reads better.

Streamflow section: the rating curve should be shown and the rating curve data could be included in the dataset as well. This may be useful if the rating curve is to be

updated with new measurements in the future (or if another function is used to create the rating curve). This is important to evaluate the uncertainties on the discharge data. I did not understand well why it was necessary to make two rating curves depending on the season. When the snow covers the stream there is no discharge measurements anyway. In addition it seems that only the summer rating curve was used? This section could be elaborated and clarified.

The landcover data section should be revised because the methodology is too vague. A list GIS software is not sufficient to understand the workflow. The authors should provide details on the algorithm and parameters (e.g. thresholds on the NDSI/NDWI, etc.). Was it done by supervised or unsupervised classification? If supervised, details on the training samples are needed. I understand that top-of-atmosphere reflectance images were transformed to albedo maps using the Liang et al method. First it is not clear to me why the albedo was used instead of the reflectance to make these land cover maps. Second the narrowband to broadband albedo should be applied to surface reflectance, not TOA reflectances. How was done the atmospheric correction?

Otherwise I agree with the comments of Referee #1.

Minor comments

Fig 1: add legend to the upper panel (blue areas = glacier) P4L5: et al. in italics Fig 5: "with 2.75 multiplier applied to shortwave radiation to mimic noon values" Clarify. Fig 16: x label cropped on the right

---

## Author Comment (AC1) · 11 Feb 2021

GENERAL COMMENT

RC1: The authors present hydrometeorological and glaciological observations of the Peyto Glacier Research Basin in Canada, including high resolution DEMs, long term meteorological data, precipitation, outflow from the glacier and bias corrected reanalysis information. In addition, the authors provide an interesting description of the historical monitoring efforts of the basin. In this respect, I found the dataset a good contribution to the already available information over mountainous regions that could asses the basis of future glaciological studies in the area. In general, I find the dataset very valuable and self-explanatory, however the description of the dataset in the manuscript is sometimes a bit confusing. Some paragraphs mix the methods to process and validate the data, and the data available from other sources, with the description of the dataset itself. My mayor concern about the dataset is the chosen platform to share the data. It does not allow simultaneous downloads, unless the user registers on it. This is highly annoying as the user have to download and install specific software from the platform. Such software comes with third party libraries that Linux users have to install manually. It seems like such libraries are not totally supported by all systems, or are not totally supported in an "user-friendly" way. Thus, it is very likely that the linux users have to use specific command line tools, just to download few Mbs of data. Furthermore, we found the platform surprisingly slow in comparison with other repositories. The authors should consider migrate the dataset to other platform, or at least submit a single file compressed version of the whole dataset that will allow future users to download the complete dataset avoiding the use of the platform GUI application and registration.

Response: We are grateful to Refereee#1 for the encouragement and positive statement. We apologize for the difficulties that they had to face downloading the dataset. As suggested, a single compressed file for the whole dataset has been uploaded on the FRDR. The README file provides details on these and options to download the whole dataset.

SPECIFIC COMMENTS:

RC1: Table1: Maybe I misunderstood something, but I can not find in the database the same variables reported in the Table 1. For instance, Peyto Main should include [Ta, RH, Ws, Wd, Ts, Qsi, Qso, Qli, Qlo, Ppt, P, Sd], but the file in the dataset includes just [Ta, RH, Ws, Qsi, Qli]. Why not to include all the variables?

Response: We have cleaned and uploaded only those variables which were recorded

long-term and which are generally used for meteorological forcing to a hydrological model. Ppt was not included for the Peyto Main as detailed in the section 3.2 Precipitation because it is unreliable. Ppt data from Bow Summit has been included instead. We have put a note on Table 1 to indicate which data are in the FRDR repository to make it clear which data were cleaned, corrected and stored.

RC1: 11p/ Lines 5 to 10. It is unclear if the data provided is the raw information or the corrected one. If it is the raw data please highlight, if not, the raw data should be included.

Response: They are corrected data. There were several errors and inconsistencies in data formats, including timing and data logger programming errors in the raw data. Corrected data were also extracted from raw data on paper. This involved several years of tedious work. For these reasons, the raw data should not be (due to errors) or cannot be (due to paper) included in a digital archive.

RC1: I miss a brief description of the followed methodology for the bias correction of the solid precipitation for wind induced undercatch.

Response: There is no correction to solid precipitation in the Peyto Main Old data file because we are not confident that the gauge correction can be successfully applied in this extremely windy and gusty environment. Also, for many years the precipitation gauge was not filled with antifreeze, methanol and oil fluids as per recommendations – this makes it unreliable. This description is now included in the metadata. Figure 7 shows this, where we applied an Alter shield undercatch correction (Smith, 2007) to compare wind undercatch corrected precipitation to that measured in a sheltered, lower elevation site at Bow Summit. This figure shows that even with correction for wind undercatch the Peyto Main Old precipitation cannot match precipitation in the valley below it and so is not reliable or useable and this is discussed on page 12, lines 6-9.

RC1: Figure 11. Is the gap filling procedure applied when the gap is bigger than 4

hours or lower than 5 (form the text)? It will be interesting to flag the filled timesteps.

Response: For gap filling, interpolation is applied when a gap is less than or equal to 4 hours. When the gap is longer than 4 hours, then a regression method is applied to infill data using nearby station data. The text on page 15, line 8 has now been rephrased to make this clearer. The revised dataset has flags to denote infilled data.

RC1: It is surprising that the authors have chosen the deprecated ERA-Interim reanalysis instead of ERA5. Is there any reason for that? I am not familiar with WFDEI, but is it not just a bias corrected version of the Era-Interim with a spatial interpolation improve the resolution? If the idea was to perform a bias correction, Why not to use just ERA5-Land reaching a much higher resolution?

Response: We used the EU-WATCH WFDEI data set as this employs a gauge-based bias correction and spatial interpolation of the ERA-Interim that vastly improves its reliability for hydrological purposes. The WATCH group has not produced a bias corrected ERA-5 product that is available for this analysis and so we believe the WFDEI is the best reanalysis product available for hydrological purposes. We were able to extrapolate this successfully to Peyto Glacier. It has also performed well in the region in other model tests.

RC1: 19p/ Lines 12-20. As is presented as forcing data, it would be interesting for users to know the elevation and coordinates of the original reanalysis cells. If after the bias correction the elevation of the reanalysis cells should be considered the same as for main station, please highlight. Why not to use other stations too?. It will be interesting to read few comments about the limitations of bias corrected reanalysis specifically in terms of resolution compared with the distributed in situ observations. In addition, the metadata of the dataset highlights that the bias correction was performed using different stations (Metadata: "Bias corrected to Peyto Main for teauQsiQli, and to BowSummit for p"). It should be specified and justified in the text.

Response: Elevation and coordinates of the original reanalysis cells are now added

in the metadata descriptions and uploaded in the revised dataset. Texts are added as suggested.

RC1: 3.6 Glaciological data. Not all this data sources are available in the link provided by the authors (e.g. repeat photography). Please clarify when you are summarizing the available information, and when you are describing the new dataset, specifically in the second paragraph. Please, use same coordinate reference systems for all the geospatial data. (e.g. BasinBoundary).

Response: We have added the descriptions and data sources to the published dataset as suggested. We have clarified in Table 1 which data are in the dataset and which are not. We now use the same reference system - WGS 84 / UTM zone 11N (EPSG:32611) for all geospatial data.

RC1: 25p/Line 10. Could you provide a description of these "stable terrain"?

Response: This terrain excludes trees, lakes/water bodies, glaciers, or snow cover. This is now more clearly defined in the text.

RC1: 26p./Line 3 to 4. the authors have probably used the tool GeoUtils, not the repository it self. I find the whole sentence confusing.

Response: All the pixels outside this mask were classified as stable terrain and used for co-registration. We used the co-registration script available in github repository at https://github.com/GeoUtils. The statistics of the elevation difference for stable terrain after co-registration are listed in Table 9. Text describing this has now been added.

RC1: 27p/Line 20. Again, this 1966 Land cover map is not available in the link provided by the authors.

Response: Thank you for pointing out this. Now the map has been added to the revised dataset.

++++++

Anonymous Referee #2

GENERAL COMMENT:

RC2: This paper presents a valuable collection of long term glaciological, hydrological and meteorological observations in a well-studied glacier-fed catchment. Although most of the datasets were already used elsewhere the authors argue that these datasets have "never been assembled in a single description until now". I am not familiar with the study region so I cannot verify this statement. However, it may be useful to emphasize which dataset was already published and which one was not (maybe in a table). The paper reads well in general, but some sections reads better.

Response: We are grateful to Refereee#2 for the encouragement. Details have been added to the revised metadata to indicate previously published data, most of which were in graphical form, or annual values, and very few in the form of readily accessible, numerical data files, certainly not with the lengths of records described here. This paper is certainly the first to comprehensively describe and present the data over long time periods for Peyto Glacier Research Basin.

SPECIFIC COMMENTS:

RC2: Streamflow section: the rating curve should be shown and the rating curve data could be included in the dataset as well. This may be useful if the rating curve is to be updated with new measurements in the future (or if another function is used to create the rating curve). This is important to evaluate the uncertainties on the discharge data. I did not understand well why it was necessary to make two rating curves depending on the season. When the snow covers the stream there is no discharge measurements anyway. In addition, it seems that only the summer rating curve was used? This section could be elaborated and clarified.

Response: The streamflow section (P.21 line 17-34) has been revised to incorporate the comment and increase clarity. the mention of the snow-cover rating curve is removed, as it was indeed not used to calculate discharge due to high uncertainty. The measure of uncertainty of the streamflow measurements is also added. A figure with the rating curve and calculated discharge is also added to the manuscript. The flow and stage data use to calculate the rating curve have been added to the revised repository.

RC2: The landcover data section should be revised because the methodology is too vague. A list GIS software is not sufficient to understand the workflow. The authors should provide details on the algorithm and parameters (e.g. thresholds on the NDSI/NDWI, etc.). Was it done by supervised or unsupervised classification? If supervised, details on the training samples are needed. I understand that top-of-atmosphere reflectance images were transformed to albedo maps using the Liang et al method. First it is not clear to me why the albedo was used instead of the reflectance to make these land cover maps. Second the narrowband to broadband albedo should be applied to surface reflectance, not TOA reflectances. How was done the atmospheric correction?

Response: The landcover data section 3.7.2 [page 27, lines 1 to 35] have been revised to incorporate the comment and increase clarity. The workflow used to determine the landcover is now defined with NDSI and NDWI values.

We have followed the use of TOA values as a standard operating procedure in this work, with appropriate narrow to broadband conversion (Hall et al., 2002; Hall and Riggs, 2007; Liang, 2000; Smith, 2010) as we appreciate the fact that atmospheric backscatter will inflate surface reflectance values, ice albedo values measured on Peyto as well as those obtained from atmosphere corrected satellite images of Peyto range from 0.17 to 0.3 (Cutler, 2006), so backscatter inflation of albedo is unlikely to reach 0.4.

Snow and firn areas within firn/snow landcover were classified by their albedo (Liang, 2000; Smith, 2010) as snow possesses higher albedo than the ice counterpart. The NDSI, NDWI and albedo for the images were obtained from the calculation on the

Google Earth Engine platform. The threshold of NDSI for snow cover is kept at $\geq 0.4$ (Hall et al., 2002; Hall and Riggs, 2007). NDWI tends to possess dynamic threshold value (Ji et al., 2009). In our case keeping the threshold to 0.4 showed best classification for a waterbody as a lower value tends to misclassify ice pixels as waterbodies. Similarly, albedo with the threshold of $\geq 0.4$ was considered to classify firn and that of less than 0.4 to classify ice within already classified NDSI based glacier area. The noise in the landcover classification was cleaned with an elimination function on ArcMap, visual inspection, manual correction of a few misclassified areas, and finally the files were clipped by the boundary of the PGRB. These have been now addressed in the revised paper. The new landcover maps from 2017 and 2018 have also been added. These are also added in Table 10.

RC2: Fig 1: add legend to the upper panel (blue areas = glacier) P4L5: et al. in italics Fig 5: "with 2.75 multiplier applied to shortwave radiation to mimic noon values" Clarify.

Response: This multiplier was used to make the display easier to see but, noting the reviewer's comment on the possibility for confusion, it has been changed to a conventional display without a multiplier.

RC2: Fig 16: x label cropped on the right

Response: Thank you for pointing out this. We have fixed it now.

+++++++

References:

Cutler, P. M.: A Reflectivity Parameterization for use in Distributed Glacier Melt Models, Based on Measurements from Peyto Glacier, Canada, in Peyto Glacier: One Century of Science, edited by M. N. Demuth, D. S. Munro, and G. J. Young, pp. 179–200, National Hydrology Research Institute Science Report 8., 2006.

Hall, D. K. and Riggs, G. A.: Accuracy assessment of the MODIS snow products, Hydrol. Process., 21(November 2008), 1534–1547, doi:10.1002/hyp, 2007.

Hall, D. K., Riggs, G. A., Salomonson, V. V, DiGirolamo, N. E. and Bayr, K. J.: MODIS snow-cover products, Remote Sens. Environ., 83(1–2), 181–194, doi:10.1016/S0034-4257(02)00095-0, 2002.

Ji, L., Zhang, L. and Wylie, B.: Problems of Dynamic NDWI Threshold and Objectives of the Study The NDWI data derived from Landsat MSS, TM, and ETM (Jain et al, Photogramm. Eng. Remote Sens., 75(11), 1307–1317 [online] Available from: https://www.ingentaconnect.com/content/asprs/pers/2009/00000075/00000011/art00004?cr 2009.

Liang, S.: Narrowband to broadband conversions of land surface albedo: I. Algorithms, Remote Sens. Environ., 76, 213–238, doi:10.1016/S0034-4257(00)00205-4, 2000.

Smith, C. D.: Correcting the Wind Bias in Snowfall Measurements Made with a Geonor T-200B Precipitation Gauge and Alter Wind Shield, in CMOS Bulletin SCMO, 36(5), 162-167., vol. 36, pp. 162–167., 2007.

Smith, R. B.: The heat budget of the earth's surface deduced from space, Yale Univ. Cent. Earth Obs., (0), 1–11 [online] Available from: https://yceo.yale.edu/sites/default/files/files/Surface_Heat_Budget_From_Space.pdf (Accessed 15 February 2018), 2010.